materials science/nanotechnology

physical vapour deposition-grown, tellurium, transistors, mobility

**Authors for correspondence:**
Bin Yao
e-mail: binyao@jlu.edu.cn
Yujue Yang
e-mail: yujue@mail.ustc.edu.cn
Nengjie Huo
e-mail: njhuo@m.scnu.edu.cn

†These authors contributed equally to this study. This article has been edited by the Royal Society of Chemistry, including the commissioning, peer review process and editorial aspects up to the point of acceptance.

# High hole mobility in physical vapour deposition-grown tellurium-based transistors

Lin Tao[1,2,†], Lixiang Han[3,†], Qian Yue[5], Bin Yao[1,2], Yujue Yang[4] and Nengjie Huo[5]

[1]State Key Lab of Superhard Material, and College of Physics, and [2]Key Laboratory of Physics and Technology for Advanced Batteries (Ministry of Education), College of Physics, Jilin University, Changchun 130012, People's Republic of China
[3]School of Materials and Energy, and [4]School of Physics and Optoelectronic Engineering, Guangdong University of Technology, Guangzhou 510006, People's Republic of China
[5]Institute of Semiconductors, South China Normal University, Guangzhou 510631, People's Republic of China

NH, 0000-0003-2520-6243

Carrier mobility is one of most important figures of merit for materials that can determine to a large extent the corresponding device performances. So far, extensive efforts have been devoted to the mobility improvement of two-dimensional (2D) materials regarded as promising candidates to complement the conventional semiconductors. Graphene has amazing mobility but suffers from zero bandgap. Subsequently, 2D transition-metal dichalcogenides benefit from their sizable bandgap while the mobility is limited. Recently, the 2D elemental materials such as the representative black phosphorus can combine the high mobility with moderate bandgap; however the air-stability is a challenge. Here, we report air-stable tellurium flakes and wires using the facile and scalable physical vapour deposition (PVD) method. The prototype field-effect transistors were fabricated to exhibit high hole mobility up to 1485 cm$^2$ V$^{-1}$ s$^{-1}$ at room temperature and 3500 cm$^2$ V$^{-1}$ s$^{-1}$ at low temperature (2 K). This work can attract numerous attentions on this new emerging 2D tellurium and open up a new way for exploring high-performance optoelectronics based on the PVD-grown p-type tellurium.

## 1. Introduction

So far, two-dimensional (2D) materials have attracted numerous attentions due to their unique physical properties such as tunable band structures, atomically thin profile, strong light–

matter interactions and mechanical flexibility, offering new fascinating possibilities in various applications including wearable electronics, photonics and logic circuits [1–3]. One of the most important figures of merit for 2D materials is their carrier mobility, which determines to a large extent their performance as electronic or optoelectronic devices. As the first discovered 2D material, graphene possesses ultrahigh mobility exceeding $10\,000\,cm^2\,V^{-1}\,s^{-1}$ at room temperature due to the linear energy dispersion [4]. This makes graphene suitable for high-speed photodetectors (up to 40 GHz bandwidth) [5], but the single atomic layer and zero bandgap lead to the large on-current and weak photo-absorption, limiting the on–off ratio of transistors and photo-responsivity of photodetectors based on graphene. Subsequently, the 2D transition-metal dichalcogenides (TMDs) emerge as promising candidates in transistors and photodetectors [6,7] benefiting from their sizable bandgap ranging from 1 eV to 2 eV. However, most TMDs were reported to exhibit n-type behaviour and the enhancement of mobility always remains a challenge. Through the utilization of high-$k$ dielectric as strongly coupled top-gate insulator, the exfoliated monolayer $MoS_2$ can exhibit room temperature mobility up to $200\,cm^2\,V^{-1}\,s^{-1}$ and on–off ratio of transistors up to $10^8$ [8]. The high mobility leads to an external photo-responsivity of $880\,A\,W^{-1}$ [9] The mobility of monolayer $WS_2$ can also reach $50–83\,cm^2\,V^{-1}\,s^{-1}$ by the contact and dielectric engineering [10,11]. The 2D $WSe_2$ was reported to exhibit ambipolar behaviour with maximum hole mobility of approximately $250\,cm^2\,V^{-1}\,s^{-1}$ in monolayer and $500\,cm^2\,V^{-1}\,s^{-1}$ in bulk [12,13], which enables highly efficient photovoltaic or light emitting devices by creating in-plane p–n junctions through local electrostatic doping [14]. Compared to exfoliated samples, chemical vapour deposition (CVD)-grown monolayer TMDs suffer from relatively low carrier mobility due to the numerous defects or grain boundaries. Significant efforts such as annealing smooth substrates [15], using metal–organic precursors [16], oxygen-assisted growth [17] and phonon suppression [18] have been devoted to optimize the carrier mobility of CVD-$MoS_2$ in the range of $20–90\,cm^2\,V^{-1}\,s^{-1}$ at room temperature.

Following graphene and TMDs, 2D semiconducting monoelemental materials have recently emerged as new class of platforms for optoelectronics. As a typical one, black phosphorus (b-P) with a tunable bandgap (0.3–2.0 eV) exhibited high hole mobility up to $1000\,cm^2\,V^{-1}\,s^{-1}$, demonstrating great potential in transistors and infrared photodetectors but suffering from its poor air-stability [19,20]. As the cousin of b-P, black arsenic (b-As) was predicted to have hole mobility of $10^3\,cm^2\,V^{-1}\,s^{-1}$ [21]. However, the maximum experimental hole mobility of few-layered b-As nanosheets is approximately $59\,cm^2\,V^{-1}\,s^{-1}$, which is much lower than the theoretical value because of the abundant defects and poor air-stability [22]. The few-layer bismuth (Bi), belonging to the same $V_A$ group, exhibited a mobility up to approximately $220\,cm^2\,V^{-1}\,s^{-1}$, which is regarded as a 2D topological insulator with non-trivial topological edge states [23,24].

Very recently, 2D tellurium has become a new promising p-type material, benefiting from its high mobility, narrow bandgap and high optical absorption as well as an extraordinary air-stability in contrast to other elemental 2D materials (b-P, b-As, Bi, etc.). This offers tellurium the great application potential in field-effect transistors (FETs) [25,26], polarized infrared imaging [27], quantum information [28] and photodetectors [27,29]. In theory, the few-layer Te possesses ultrahigh theoretical hole mobility ($10^4–10^6\,cm^2\,V^{-1}\,s^{-1}$) due to the wave function hybridization for covalent-like quasi-bonding [30]. Subsequently, the solution-synthesized and air-stable quasi-2D tellurium (Te) nanoflakes with bandgap of 0.31 eV for short-wave infrared photodetectors were fabricated to exhibit room temperature hole mobilities of $450\,cm^2\,V^{-1}\,s^{-1}$ and photo-responsivity from $13\,A\,W^{-1}$ (1.4 μm) to $8\,A\,W^{-1}$ (2.4 μm) [27]. Then the thickness-dependent 2D Te FETs were further optimized to exhibit the reported hole mobility of $700\,cm^2\,V^{-1}\,s^{-1}$, which were also prepared by the solution exfoliation method [25]. To promote the large-scale processing, Javey et al. prepared wafer-scale polycrystalline Te FETs, logic gates and computational circuit by low temperature deposition method [26]. In results, the obtained p-type Te FETs exhibited an effective hole mobility of approximately $35\,cm^2\,V^{-1}\,s^{-1}$, on–off current ratio of approximately $10^4$ and subthreshold swing of $108\,mV\,dec^{-1}$. Meanwhile, high-quality 2D tellurene thin films were acquired from the hydrothermal method with high hole mobility of nearly $3000\,cm^2\,V^{-1}\,s^{-1}$ at low temperatures (0.2 K), which allows the observation of well-developed Shubnikov–de-Haas oscillations and quantum Hall effect (QHE) [28].

Thus, the fascinating physical properties, particularly the high hole mobility, have promoted the 2D tellurium a star in materials and physics field. However, the reported highest mobility is still far below the theoretical value. In this work, we prepared tellurium flakes and wires using the physical vapour deposition (PVD) method which is more facile and scalable compared to the reported solution-based or low temperature deposition method. The prototype FETs based on the obtained Te flakes were fabricated to exhibit p-type behaviour with field-effect mobility up to approximately $900\,cm^2\,V^{-1}\,s^{-1}$

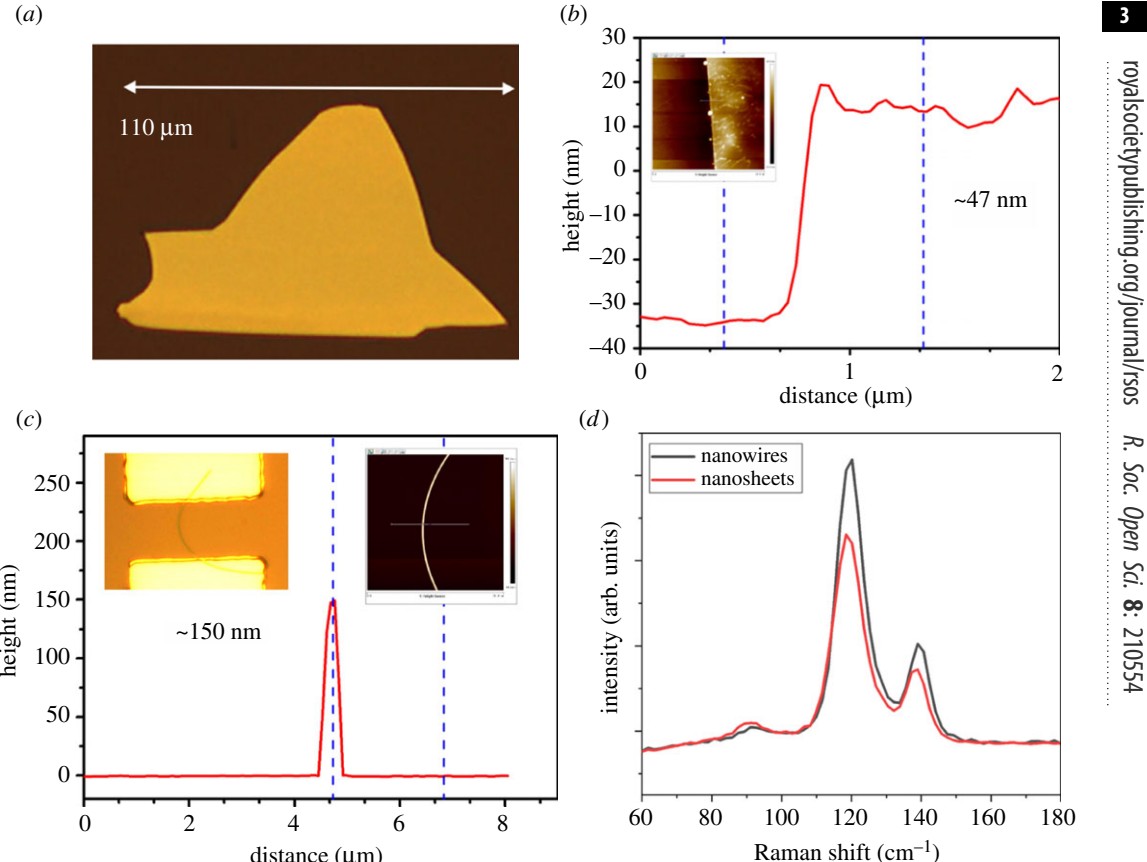

**Figure 1.** (*a*) OM image of PVD-grown Te nanosheet. (*b*) Height profile of Te nanosheet in (*a*); inset is the AFM image. (*c*) Height profile of Te nanowire; inset images are optical and AFM images. (*d*) Raman spectra of Te nanowire and nanosheet.

and Hall mobility as high as 1485 cm$^2$ V$^{-1}$ s$^{-1}$ at room temperature, which is a record value so far and even outperforms the conventional p-type silicon or germanium. This work refreshed the world record value in hole mobility of tellurium, and could attract numerous attentions for further exploration in novel physics and device applications such as the QHE or acting as p-type components in optoelectronics.

## 2. Results and discussion

The tellurium was grown directly on the SiO$_2$/Si substrate using the PVD method, the details for the growth procedure can be found in §4. Figure 1*a* shows the optical microscopy (OM) image of Te flakes with large size of 110 µm. Figure 1*b* is the corresponding height profile of Te flakes showing the thickness of approximately 47 nm. The curved tellurium wire with diameter of 150 nm is shown in figure 1*c*. The Raman spectra were also obtained as shown in figure 1*d*, presenting the typical phonon vibration mode of tellurium, which is consistent with a previous report [26]. The SEM image of the mixture of Te sheet and wire is shown in electronic supplementary material, figure S1*a*.

Then the FETs based on both Te flakes and wires were fabricated using the lithography and metal evaporation technique. The inset of figure 2*a* shows the OM image of the FETs. The metal Ti (5 nm)/ Au (50 nm) was deposited as source and drain electrodes; the bottom SiO$_2$ (285 nm) and heavily doped Si acted as gate insulator and gate electrodes, respectively. The length and width of channel are 34 µm and 12 µm, respectively. The thickness is approximately 230 nm from the atomic force microscopy (AFM) image (electronic supplementary material, figure S2). After annealing the devices under vacuum at 100°C for 2 h to improve the contact quality, the electrical properties were measured at room temperature and ambient environment. The transfer and output characteristics are shown in figure 2*a* and electronic supplementary material, figure S1*b*, respectively. Clearly, the Te nanosheets exhibit p-type behaviour with current on–off ratio of approximately 70. The carrier mobility can be

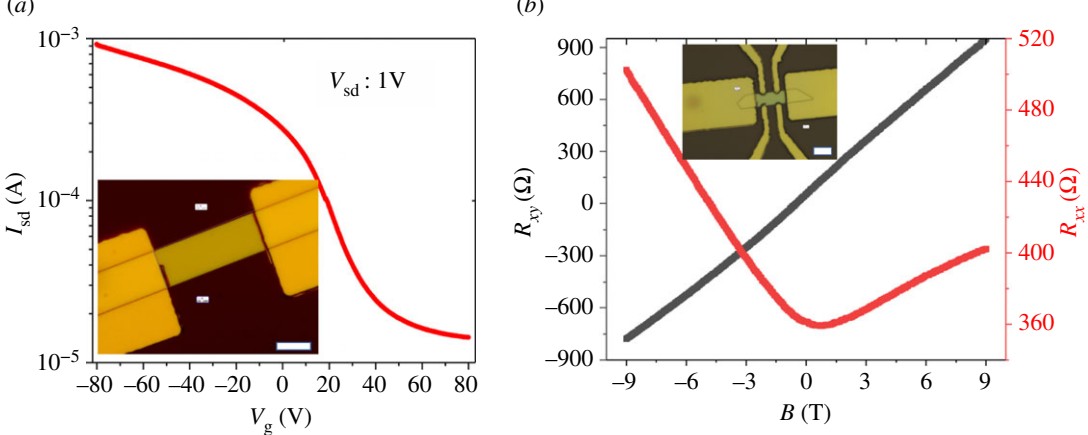

**Figure 2.** (*a*) Transfer curves of Te nanosheets-based FETs. The inset is the OM image of the device. (*b*) Hall transport measurement of the Te nanosheets. The inset is the OM image of the device with Hall bar geometry. The scale bar in the inset is 10 μm.

calculated using the formula: $\mu = \partial I_{DS}/\partial V_G(L/WC_iV_{DS})$, where $L$ is the channel length, $W$ is the channel width, and $C_i$ is the gate capacitance between the channel and the silicon back gate per unit area, which can be given by $C_i = \varepsilon_0\varepsilon_r/d$, where $\varepsilon_0$ is vacuum dielectric constant and $\varepsilon_r$ (3.9) and $d$ (285 nm) are dielectric constant and thickness of $SiO_2$, respectively. Thus, the hole mobility at room temperature is calculated to be as high as $901\ cm^2\ V^{-1}\ s^{-1}$. Our PVD-grown Te flakes even outperform the conventional semiconductors such as the P-doped silicon and germanium in terms of the hole mobility. It is noted that the mobility can retain the same level after placing the devices in ambient environment for two months, implying the high air-stability. The forward and backward transfer curves were almost overlapped as shown in electronic supplementary material, figure S1*c*, which is attributed to the fewer charge trap states in the high-quality samples.

The Hall bar devices based on the Te flakes were also fabricated on the $SiO_2$/Si substrate shown in inset of figure 2*b*. The length and width of channel are 19 μm and 8 μm, respectively. Then the Hall measurement was performed using the PPMS systems at both room and low temperature. Figure 2*b* shows the longitudinal resistance ($R_{xx}$) and transverse resistance ($R_{xy}$) as functions of applied magnetic field at 2 K, indicating the well positive magnetoresistance effect and large Hall coefficient. The Hall mobility can be calculated using the following formula: $\mu = \sigma C_H = \sigma(R_H/B) \times d$, where $C_H$ and $R_H$ are the Hall coefficient and transverse resistance ($R_{xy}$), respectively, $B$ is the applied magnetic field, $d$ is the thickness of sample, and $\sigma$ is the conductivity. As a result, we achieved a Hall hole mobility up to $1485\ cm^2\ V^{-1}\ s^{-1}$ at 300 K and $3500\ cm^2\ V^{-1}\ s^{-1}$ at 2 K, outperforming most 2D materials and conventional P-doped semiconductors. The higher Hall mobility compared to the field-effect mobility is due to the removal of the influence from the contact barrier and the surface adsorbates such as $H_2O$ or $O_2$ in high vacuum chamber during the Hall measurement. These adsorbates can act as scattering centres to reduce the mobility. The FETs based on the thinner Te flakes with thickness of 47 nm were also fabricated; however, the performances were relatively poor compared to that in the thicker samples (electronic supplementary material, figure S3), which was probably attributed to the poor quality of the thin samples and the more obvious influence of charge scattering from the strap states between sample and substrate.

We also fabricated the FETs based on the Te wires as shown in the inset of figure 3*a*. The field-effect characteristics were measured in air at room temperature. Figure 3*a* and electronic supplementary material, figure S1*d*, show the transfer and output curves, respectively. We observed that the Te wires also exhibited p-type behaviour. The hole mobility was calculated to be $833\ cm^2\ V^{-1}\ s^{-1}$ at room temperature, which is smaller than that in Te flakes. The current hysteresis in the transfer curves shown in figure 3*b* indicates the existence of trap states which can act as scattering centres to influence the mobility in wires. On the other hand, the mobility is still much higher than that of other reported one-dimensional (1D) nanowires such as ZnO, CdSe, $Bi_2S_3$, etc. Our 1D nanowires with high hole mobility could be applied in some special application scenarios such as 1D piezotronics and 1D–2D mixed-dimensional optoelectronics.

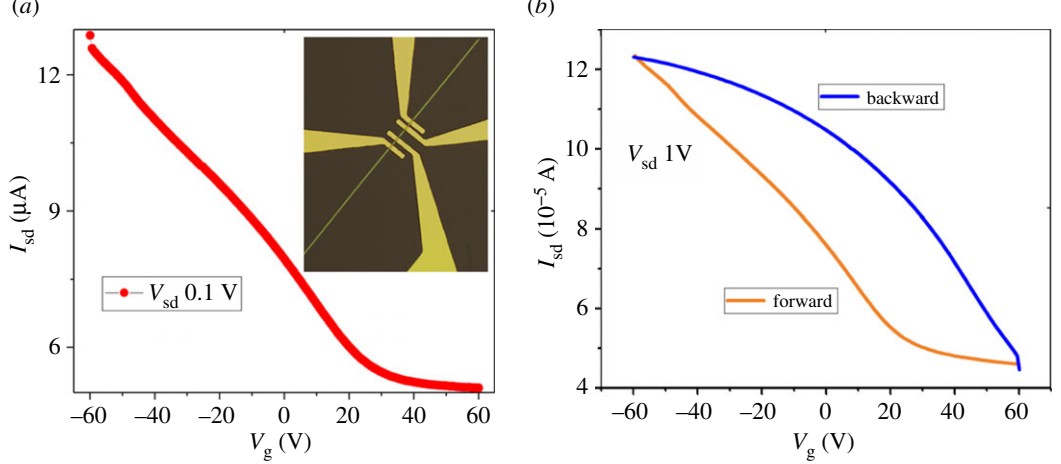

**Figure 3.** (*a*) Transfer curves of the Te wires-based FETs. The inset is the OM image of the device. (*b*) The transfer curves with forward and backward sweeping of the gate voltage showing the hysteresis behaviour.

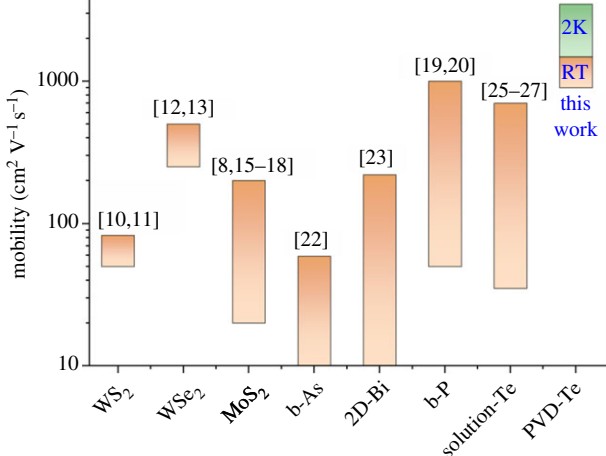

**Figure 4.** Comparison of mobility among different kinds of 2D materials, showing the superior mobility up to 1485 cm$^2$ V$^{-1}$ s$^{-1}$ (room temperature, RT) and 3500 cm$^2$ V$^{-1}$ s$^{-1}$ (2 K) in our PVD-grown Te flakes with thickness of 230 nm.

## 3. Conclusion

In summary, the tellurium flakes and wires were successfully prepared using the facile and scalable PVD method. We achieved the record value in hole mobility up to 1485 cm$^2$ V$^{-1}$ s$^{-1}$ in Te flakes with thickness of 230 nm at room temperature, outperforming most 2D materials such as TMDs, Bi, b-P, b-As as well as the Te prepared by previous solution method as shown in figure 4. The Te wires also exhibit high hole mobility of 833 cm$^2$ V$^{-1}$ s$^{-1}$ and offer a new possibility for mixed-dimensional optoelectronics. This work reports the p-type PVD-grown tellurium with high carrier mobility, and can pave the way towards the use of high-quality PVD-grown tellurium in unique optoelectronics including FETs, photodetectors and photovoltaic cells.

## 4. Experimental procedure

High-quality mixtures of Te flakes and wires were grown in a tube furnace using the PVD method under ambient condition and some of them were grown in tilted form. In brief, high-purity Te powder (99.999%) was first put in the centre of the quartz tube, and then placed in the centre of the furnace. The SiO$_2$/Si substrate was placed in the downstream area. The quartz tube was sealed and flushed for 5 min using a gas of hydrogen (4%) argon mixture under 600 sccm to provide oxygen-free

environment. Then, the mixed gas was turned off during the heating up process. When the furnace was heated up to 500°C within 8 min, the mixed gas was turned on under 500 sccm to carry high density vapour of Te to the substrate and maintained for 3 min. The gas was turned off immediately once the growth was finished. The furnace cover was lifted for fast cooling.

Data accessibility. SEM, AFM and electrical properties of Te flakes with thinner thickness can be found in electronic supplementary material.

Authors' contributions. L.T. carried out the experimental work. L.H. participated in data analysis. Y.Y. participated in the design of the study and drafted the manuscript. Q.Y. participated in data analysis and figure plotting. B.Y. conceived of the study. N.H. designed the study and drafted the manuscript. All authors gave final approval for publication.

Competing interests. We declare we have no competing interests.

Funding. This work is supported by the 'Pearl River Talent Recruitment Program' (grant no. 2019ZT08X639) and the National Natural Science Foundation of China (grant nos. 11904108 and 61805045). This work is also supported by the National Natural Science Foundation of China under grant no. 61774075, the Science and Technology Development Project of Jilin Province under grant no. 20170101142JC, Natural Science Foundation of Jilin Province under grant no. 20180101227JC. This work was also supported by High Performance Computing Center of Jilin University, China.

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
