## [Peer Review File · Royal Society Open Science]

Review History

RSOS-202180.R0 (Original submission)

Review form: Reviewer 1

Is the manuscript scientifically sound in its present form?

No

Are the interpretations and conclusions justified by the results?

No

Is the language acceptable?

Yes

Do you have any ethical concerns with this paper?

No

Have you any concerns about statistical analyses in this paper?

No

Recommendation?

Reject

Comments to the Author(s)

Please see the attached files (see Appendix A).

Review form: Reviewer 2

Is the manuscript scientifically sound in its present form?

No

Are the interpretations and conclusions justified by the results?

No

Is the language acceptable?

Yes

Do you have any ethical concerns with this paper?

No

Have you any concerns about statistical analyses in this paper?

No

Recommendation?

Major revision is needed (please make suggestions in comments)

Comments to the Author(s)

The present manuscript reports Ultrahigh hole mobility in PVD-grown tellurium-based transistors. In my opinion, the manuscript is suitable for publication in Royal Society Open Science, after the authors have addressed the following comments and questions:

1. The author state the synthesis of Te nanowires and nanosheets, but the diameter of the nanowires around $1\mu\text{m}$ and length of $120\mu\text{m}$ is not in the nanorange. Also the OM image shows Te nanosheet with a wire attached to it. It will be nice to give additional information whether the synthesis approach used in the present work yields nanowires or nanosheets or a mixture of both.
2. A detailed TEM of the as grown nanostructure to be included just to give a clear understanding of the crystal formation.
3. The experimental section is not written completely.

Overall, this is a good paper. The technological advance to me is quite nice.

Decision letter (RSOS-202180.R0)

Dear Professor Huo:

Manuscript ID: RSOS-202180

Title: "Ultrahigh hole mobility in PVD-grown tellurium-based transistors"

Thank you for submitting the above manuscript to Royal Society Open Science. Your paper was sent to reviewers and their comments are included at the bottom of this letter.

In view of the concerns raised by the reviewers, the manuscript has been rejected in its current form. However, a new manuscript may be submitted which takes into consideration these comments.

Please note that resubmitting your manuscript does not guarantee eventual acceptance, and that your resubmission will be subject to peer review before a decision is made.

Your resubmitted manuscript should be submitted by 10-Aug-2021. If you are unable to submit by this date please contact the Editorial Office.

On behalf of the Subject Editor Professor Anthony Stace and the Associate Editor Dr Dattatray Late

REVIEWER(S) REPORTS:
Associate Editor Comments to Author ():
RSC Associate Editor:
Comments to the Author:
Reject & allow resubmission

RSC Subject Editor:
Comments to the Author:
(There are no comments.)

Reviewers' Comments to Author:
Reviewer: 1
Comments to the Author(s)
Please see the attached files.

Reviewer: 2

Comments to the Author(s)

The present manuscript reports Ultrahigh hole mobility in PVD-grown tellurium-based transistors. In my opinion, the manuscript is suitable for publication in Royal Society Open Science, after the authors have addressed the following comments and questions:

1. The author state the synthesis of Te nanowires and nanosheets, but the diameter of the nanowires around $1\mu\text{m}$ and length of $120\mu\text{m}$ is not in the nanorange. Also the OM image shows Te nanosheet with a wire attached to it. It will be nice to give additional information whether the synthesis approach used in the present work yields nanowires or nanosheets or a mixture of both.
2. A detailed TEM of the as grown nanostructure to be included just to give a clear understanding of the crystal formation.
3. The experimental section is not written completely.

Overall, this is a good paper. The technological advance to me is quite nice.

Author's Response to Decision Letter for (RSOS-202180.R0)

See Appendix B.

RSOS-210554.R0

Review form: Reviewer 2

Is the manuscript scientifically sound in its present form?

Yes

Are the interpretations and conclusions justified by the results?

Yes

Is the language acceptable?

Yes

Do you have any ethical concerns with this paper?

No

Have you any concerns about statistical analyses in this paper?

No

Recommendation?

Accept as is

Comments to the Author(s)

It is interesting to see that the present work "High hole mobility in PVD-grown tellurium-based transistors" gives excellent air stability of the as grown Te nanoflakes. However it is strongly suggested that the author should work on the improvement of the threshold and on-off ratio of the as grown nanoflakes.

Review form: Reviewer 3

Is the manuscript scientifically sound in its present form?

Yes

Are the interpretations and conclusions justified by the results?

Yes

Is the language acceptable?

Yes

Do you have any ethical concerns with this paper?

No

Have you any concerns about statistical analyses in this paper?

No

Recommendation?

Accept as is

Comments to the Author(s)

This is an interesting paper introducing the PVD synthesis of Te, which is with high mobility and is a promising 2D candidate for future application. Although the synthesis method still has a large room to be improved in terms of the thickness of sample and the poor on-off ratio of the device, the manuscript provided a simple and large scale method to synthesize Te. The revised manuscript answered most of the reviewers' comment and added some explanations to demonstrate the work more clearly. I recommend this manuscript to be accepted.

Decision letter (RSOS-210554.R0)

Dear Professor Huo:

Title: High hole mobility in PVD-grown tellurium-based transistors

Manuscript ID: RSOS-210554

It is a pleasure to accept your manuscript in its current form for publication in Royal Society Open Science. The chemistry content of Royal Society Open Science is published in collaboration with the Royal Society of Chemistry.

Please see the Royal Society Publishing guidance on how you may share your accepted author manuscript at <https://royalsociety.org/journals/ethics-policies/media-embargo/>. After publication, some additional ways to effectively promote your article can also be found here

<https://royalsociety.org/blog/2020/07/promoting-your-latest-paper-and-tracking-your-results/>.

On behalf of the Subject Editor Professor Anthony Stace and the Associate Editor Dr Dattatray Late.

RSC Associate Editor
Comments to the Author:
Accept as is

Reviewer(s)' Comments to Author:
Reviewer: 2

Comments to the Author(s)

It is interesting to see that the present work "High hole mobility in PVD-grown tellurium-based transistors" gives excellent air stability of the as grown Te nanoflakes. However it is strongly suggested that the author should work on the improvement of the threshold and on-off ratio of the as grown nanoflakes.

Reviewer: 3

Comments to the Author(s)

This is an interesting paper introducing the PVD synthesis of Te, which is with high mobility and is a promising 2D candidate for future application. Although the synthesis method still has a large room to be improved in terms of the thickness of sample and the poor on-off ratio of the device, the manuscript provided a simple and large scale method to synthesize Te. The revised manuscript answered most of the reviewers' comment and added some explanations to demonstrate the work more clearly. I recommend this manuscript to be accepted.

Appendix A

The author reported the synthesis of tellurium nanosheets and nanowires using the physical vapor deposition (PVD) method. The prototype field effect transistors (FETs) were fabricated, producing a high hole mobility at room temperature. I think the experimental data in this work is too simple and monotonous and it is not fair to compare the mobility without consideration of the thickness.

1. The diameter of “nanowire” and the thickness of “nanosheet” are 1 μm , 230 nm, respectively. In my opinion, this can not be called as “nanowire”, “nanosheet”, it is as thick as bulk materials and far from the requirements of nano-level.
2. The current on-off ratio of Te-based FET is too small, less than 100. I strongly recommend the author should grow more thinner Te plates range from 1-50 nm, and then fabricate the corresponding FETs.
3. As I mentioned before, it is too unfair to compare the mobility of Te synthesized in this work with the other published works without consideration of the thickness.
4. The mobility is not the only evaluation criterion of a FET. The on-off ratio, threshold voltage, subthreshold swing and current density are also important.
5. The experimental data in this work is too simple and monotonous, maybe the environmental stability is an interesting point, which deserved more attention and more relative studies.

Appendix B

Response to Reviewer

Reviewer 1#

Comment: The author reported the synthesis of tellurium nanosheets and nanowires using the physical vapor deposition (PVD) method. The prototype field effect transistors (FETs) were fabricated, producing a high hole mobility at room temperature. I think the experimental data in this work is too simple and monotonous and it is not fair to compare the mobility without consideration of the thickness.

Answer: Thank you so much for your professional suggestions. According to your suggestions, we have performed additional experiments to grow thinner Te nanosheets and make the corresponding field effect transistors. But we found that the thinner Te samples exhibit poor performances in terms of mobility. Thus, we still focus on the thick samples in revised manuscript. As you proposed, it is not appropriate to describe the PVD-grown Te as “nanosheet” or “nanowires” due to it is as thick as bulk materials. Thus, we re-describe the samples as Te flakes and wires in revised manuscript. We think the high mobility in bulk Te flakes grown by facile and scalable PVD method could be an important progress and we appreciate so much that you would like to recommend our work for publication in “Royal Society Open Science”.

Question 1 : The diameter of “nanowire” and the thickness of “nanosheet” are 1 μm , 230 nm , respectively. In my opinion, this cannot be called as “nanowire”, “nanosheet”, it is as thick as bulk materials and far from the requirements of nano-level.

Answer 1: Thank you for pointing out this issue. We have indeed focused on the Te samples with with large diameter or thickness because of their high carrier mobility. We also obtained the Te flakes or wires with smaller thickness and diameters. In revised manuscript, Figure 1a, b shows the OM and AFM image of Te flakes with thickness of ~ 47 nm. Figure 1c shows the AFM image of Te wires with diameter of ~ 150 nm. The corresponding FETs based on them were also fabricated as can be seen in revised

supporting Information (Figure S3 and S4). However, the performances were relatively poor compared to that in the thicker samples. The mobility of Te flakes with thickness of 47 nm is 187 cm²/Vs and the on/off ratio of Te wires is only ~2.5. The poor performances were probably attributed to the poor quality of the samples and the more obvious influence of charge scattering from the strap states between sample and substrate on the thinner samples.

Question 2: The current on-off ratio of Te-based FET is too small, less than 100. I strongly recommend the author should grow more thinner Te plates range from 1-50 nm, and then fabricate the corresponding FETs.

Answer 2: Thank you for your suggestion. We have fabricated the FETs based on the Te flakes with thickness of ~47 nm. However, from the transfer curves shown in Figure S3, the mobility is calculated to be 187 cm²/Vs and the on/off ratio is much smaller. The poor performances were probably attributed to the poor quality of the thin samples and the more obvious influence of charge scattering from adsorbates and trap states on the carrier transport. Thus, we still focused on the bulk Te flakes and their high mobility. We think the high mobility in Te flakes grown by facile and scalable PVD method could attract interests from the readers of “Royal Society Open Science”.

Questions 3: As I mentioned before, it is too unfair to compare the mobility of Te synthesized in this work with the other published works without consideration of the thickness.

Answer 3: In the revised manuscript, we have noted the thickness of the tellurium when we discussed about mobility and compared with previous work.

Question 4: The mobility is not the only evaluation criterion of a FET. The on-off ratio, threshold voltage, subthreshold swing and current density are also important.

Answer 4: Thank you for your suggestion. We agree the importance of the abovementioned characteristics of a FET. In our Te transistors, the on-off ratio is not

large compared to that in conventional Si or 2D-TMDs based transistors. As we know that the carrier mobility is one of important figures of merit for materials, that can determine to large extent the corresponding device performances. Thus, the main topic of our work is the improvement of the mobility of Tellurium as well as the developed facile and scalable PVD growth method.

Question 5: The experimental data in this work is too simple and monotonous, maybe the environmental stability is an interesting point, which deserved more attention and more relative studies.

Answer 5: We appreciate so much your suggestion. Actually, the mobility shows no obvious degradation after 2 months stored in ambient condition, indicating the excellent air stability. We have highlighted this statement. Thanks very much again.

Reviewer: 2

Comments to the Author(s)

The present manuscript reports Ultrahigh hole mobility in PVD-grown tellurium-based transistors. In my opinion, the manuscript is suitable for publication in Royal Society Open Science, after the authors have addressed the following comments and questions:

Question 1: The author state the synthesis of Te nanowires and nanosheets, but the diameter of the nanowires around 1 μ m and length of 120 μ m is not in the nanorange. Also the OM image shows Te nanosheet with a wire attached to it. It will be nice to give additional information whether the synthesis approach used in the present work yields nanowires or nanosheets or a mixture of both.

Answer 1: Thank you so much for you kind and professional suggestion. As you mentioned, it is not appropriate to describe the PVD-grown Te as “nanosheet” or “nanowires” due to it is as thick as bulk materials. Thus, we re-describe the samples as Te flakes and wires in revised manuscript. Actually, we also obtained the thinner Te flakes with thinness of 47 nm and Te wires with diameter of 150nm, but the mobility is relatively poor compared to that in thicker samples, this is probably due to the poor quality of the thin samples and the more obvious influence of charge scattering on the thin samples.

In this work, we utilized the facile and scalable PVD method to grow Te, it could yield both flakes and wires even a mixture of both in one batch. It is interesting as this facile method can produce the Te samples with different morphology and high mobility, and the underlying growth mechanism for Te is worth further studying in future.

Question 2: A detailed TEM of the as grown nanostructure to be included just to give a clear understanding of the crystal formation.

Answer 2: Thank you for your suggestion, the TEM could provide more information, while our samples are too thick exceeding 50 nm that the electron beam cannot pass through the samples, thus we cannot obtain the clear TEM pattern. We sorry about that but this may not affect the main topic of this work.

Question 3: The experimental section is not written completely.

Answer 3: Thank you for pointing out this issue. We have added the whole experimental process in Experimental section. Please check it in revised manuscript.